# A humanized minipig model for the toxicological testing of therapeutic recombinant antibodies

Tatiana Flisikowska [1,5], Jerome Egli[2,5], Krzysztof Flisikowski[1], Marlene Stumbaum[1], Erich Küng[2], Martin Ebeling[2], Roland Schmucki[2], Guy Georges[3], Thomas Singer[2], Mayuko Kurome[4], Barbara Kessler[4], Valeri Zakhartchenko[4], Eckhard Wolf[4], Felix Weber [2] ✉, Angelika Schnieke [1,5] ✉ and Antonio Iglesias[2,5]

The safety of most human recombinant proteins can be evaluated in transgenic mice tolerant to specific human proteins. However, owing to insufficient genetic diversity and to fundamental differences in immune mechanisms, small-animal models of human diseases are often unsuitable for immunogenicity testing and for predicting adverse outcomes in human patients. Most human therapeutic antibodies trigger xenogeneic responses in wild-type animals and thus rapid clearance of the drugs, which makes in vivo toxicological testing of human antibodies challenging. Here we report the generation of Göttingen minipigs carrying a mini-repertoire of human genes for the immunoglobulin heavy chains γ1 and γ4 and the immunoglobulin light chain κ. In line with observations in human patients, the genetically modified minipigs tolerated the clinically non-immunogenic IgG1κ-isotype monoclonal antibodies daratumumab and bevacizumab, and elicited antibodies against the checkpoint inhibitor atezolizumab and the engineered interleukin cergutuzumab amunaleukin. The humanized minipigs can facilitate the safety and efficacy testing of therapeutic antibodies.

The efficacy and safety of monoclonal antibodies (mAbs) can be compromised if the administration of a compound evokes an immune response in humans, which is manifested by the development of anti-drug antibodies (ADAs). Several factors can induce ADAs and are classified as related to the patient, the disease, the product or the treatment. They can lead to mild-to-life-threatening symptoms that are well-documented for several clinically approved therapeutic proteins[1]. These adverse reactions vary between compounds and are difficult to predict. There are still major gaps in understanding the pharmacokinetics of ADAs, their neutralizing ability and their cross-reactivity with endogenous molecules or other biological compounds.

To address these drawbacks, several different preclinical in silico and in vitro models have been developed[2–7]. Assessment of immunogenicity in in vivo models includes animal trials within the context of an intact immune system. However, any human protein will probably cause an immune response in a test animal because of species differences.

[1]Department of Molecular Life Sciences, Chair of Livestock Biotechnology, School of Life Sciences Weihenstephan, Technical University München, Freising, Germany. [2]Roche Pharmaceutical Research and Early Development, Roche Innovation Center Basel, Basel, Switzerland. [3]Roche Pharmaceutical Research and Early Development, Roche Innovation Center Munich, Penzberg, Germany. [4]Gene Center and Department of Veterinary Sciences, Molecular Animal Breeding and Biotechnology, Ludwig-Maximilians-Universität München, Munich, Germany. [5]These authors contributed equally: Tatiana Flisikowska, Jerome Egli, Angelika Schnieke and Antonio Iglesias. ✉e-mail: felix.weber@roche.com; schnieke@tum.de

**Fig. 1 | Constructs structure of unrearranged human immunoglobulin mini-loci. a**, An overview of the project showing the generation and application of human IgG transgenic minipigs. **b**, The secreted form of the immunoglobulin heavy (IGH-γ1-γ4) chain and **c**, the κ light chain (IGK). Figure created with BioRender.com.

This can be circumvented by using surrogate antibodies specific for the animal species (but their predictive value can be questioned) or transgenic animals that express the human protein and therefore recognize it as 'self'. Any immune response raised will therefore be due to the altered state of the recombinant protein. To avoid the generation of separate transgenic animal lines for individual therapeutic proteins, a few research groups including ours have generated transgenic mice expressing sets of human immunoglobulin genes[8–10]. In contrast to most antibody (Ab)-humanized mouse models for Ab discovery, our transgenic animals still express their endogenous immunoglobulin genes (Ig) and are thus fully immune-competent. The function of the human Ig transgenes is solely to induce tolerance and they do not necessarily need to be involved in an immune response. Therefore, the mouse lines established by us carry a human IgG1 mini-repertoire composed of only the secreted form of human Ig-γ1 heavy, as well as Ig-κ and Ig-λ light chains. The genes included are those most commonly used in humans as well as in the production of therapeutic antibodies[11]. These transgenic mice permit immunogenicity studies for a whole category of human therapeutic antibodies and the assessment of potentially immunogenic modifications in Ab preparations[10]. However, for obvious anatomical and lifespan reasons, data obtained in mice are not always directly translatable to humans in terms of application routes, pharmacokinetics and long-term toxicological assessments[12]. The International Council for Harmonization (ICH) of Technical Requirements for Registration of Pharmaceuticals for Human Use requires preclinical safety testing in one rodent species and one non-rodent species. In the case of therapeutic antibodies, non-human primates (NHPs) are often the only option. The availability of a universal non-rodent, non-NHP model for the assessment of possible immunogenic and immunotoxic properties of human recombinant Abs would be a valuable tool. It would enhance preclinical safety for this rapidly growing market.

The pig has many advantages for preclinical studies, being similar to humans in size, in the anatomy of many organ systems, and in its physiological and pathophysiological responses. Pigs have a relatively short gestation time, large litter size, rapid maturation and ease of housing under specific pathogen-free conditions. Importantly, immunological similarities to humans make the pig an ideal model for preclinical research[13–15]. As with mice, genetic engineering methods are now well-established for pigs.

## Results

### Characterization of genetically engineered minipigs

On the basis of the previous mouse results[10], transgenic Göttingen minipigs carrying a similar repertoire of human *Ig heavy* (*IGH*) and *Ig κ light* (*IGK*) chain genes were generated. The outline of the project is presented in Fig. 1a. Two expression vectors were generated. The first construct includes unrearranged germline *IGH* gene segments encoding 5 variable (V), 5 diversity (D) and 6 joining (J) elements in combination with the required sequence for driving the production of secretory human IgG1 (hIgG1) H chains. To assess whether not only hIgG1 but also the hIgG4 isotype can be expressed, the constant γ4 (Cγ4) region and corresponding switch sequences (Sμ, Iγ-Sγ) were also included in this construct (Fig. 1b, IGH-γ1-γ4; NCBI accession number: OL809665). Human IgG1 and hIgG4 were chosen as these are the two Ig isotypes most commonly used in human therapeutic mAbs[16]. The IGH-γ1-γ4 construct should permit Ig isotype switch from Cγ1 into Cγ4. The second construct contains *IGK* gene segments encoding 2V and 5J elements plus the sequence coding for the κ constant (C) region (Fig. 1c, IGK; NCBI accession number: OL809666). Upon proper rearrangement, these gene elements should generate a repertoire of human soluble IgG proteins without interfering with the process of rearrangement and repertoire formation of endogenous porcine Ig proteins as this requires the expression of the membrane-bound human Ig, which is not included in the transgenic construct (see Fig. 1b). All the V genes used in the constructs were chosen on the basis of their predominant usage in human peripheral blood[11] and were shown in the transgenic mouse model to provide a broad tolerance to human IgG1 Abs[10].

Kidney fibroblasts isolated from male Göttingen minipigs were co-transfected with the IGH-γ1-γ4 (31.4 kb) and IGK (10.9 kb) expression vectors and a selectable marker gene (phosphoglycerate kinase (PGK)-driven *blasticidin*, BS; 1.05 kb). Single-cell clones were isolated and screened by PCR for the presence of all three transgenes. Four to five cell clones were pooled and used for somatic cell nuclear transfer resulting in the birth of eight live-born male Göttingen piglets. All were

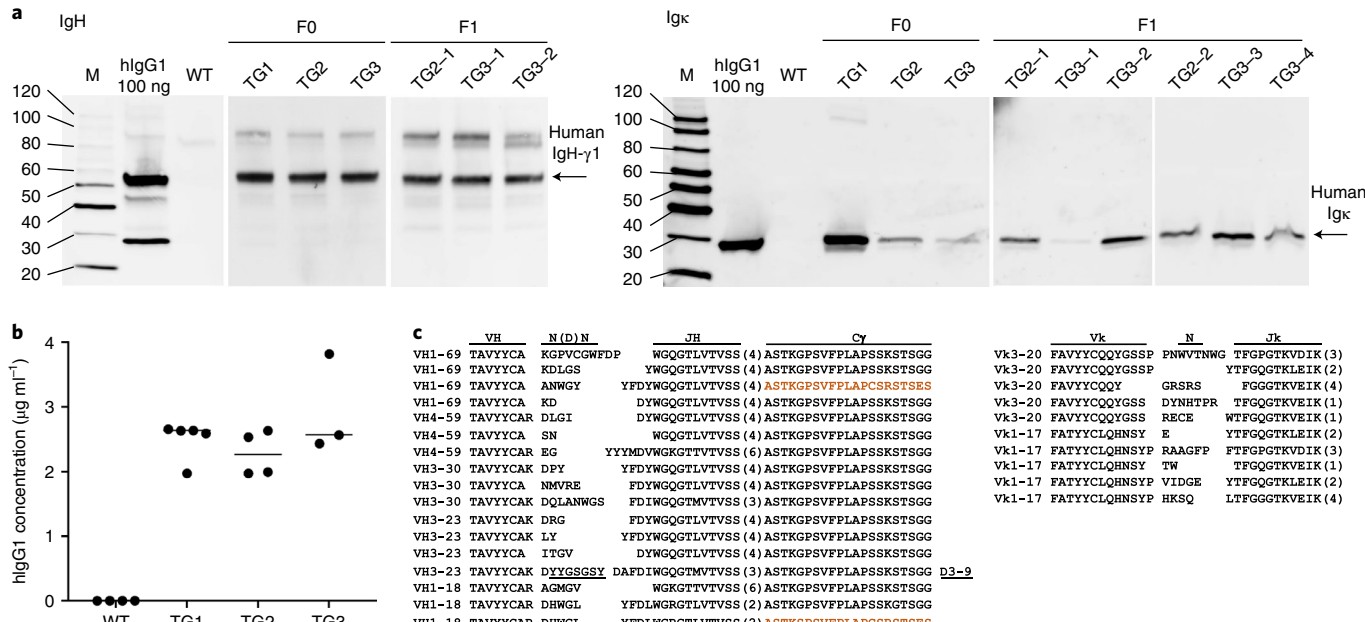

**Fig. 2 | Expression of human IgG1 protein in transgenic minipigs. a**, Western blot showing expression of IgG heavy γ1 (IgH-γ1) and κ light (IgL-κ) proteins in serum isolated from the transgenic founder (TG1–-TG3) IgG minipigs and F1 animals from line TG2 and TG3. WT Göttingen minipigs were negative for the presence of both proteins. Human IgG protein and PBS were used as positive and negative control, respectively. Unprocessed images of all western blots are shown in Supplementary Data 1. **b**, The presence of human IgG1 protein was further confirmed by ELISA analysis in serum isolated from F1 minipigs representing all 3 lines (TG line 1, $n = 5$; TG line 2, $n = 4$; TG line 3, $n = 3$). No human IgG1 was detected in WT animals ($n = 4$). Horizontal bar represents median. Each data point represents a biological replicate. **c**, Human *IgG* heavy and κ light chain genes undergo functional gene rearrangements. A selection of rearrangements of all four VH and the two Vκ transgenes is depicted. Human $V_H$- and $V_κ$-gene rearrangements show N nucleotide additions. Human IgG4 sequence of isotype switch variants is shown in brown. The J elements in the rearranged V genes are given in parenthesis, identified D gene elements are underlined and indicated at right.

positive for the presence of the three transgenes (not shown). Four founder animals reached sexual maturity (TG1–TG4). All four carried 2 copies of human IGH-γ1-γ4 and IGK transgene as determined by droplet PCR but differ in the copy numbers for the selectable marker gene (TG1–TG3, 1 copy; TG4, 3 copies). Except for TG4, all founder animals expressed the human IgG heavy (HC) and light (LC) chain (Fig. 2). Consequently, only TG1–TG3 were used for further breeding. All offspring (F1–F3) exhibited Mendelian transgenes inheritance, indicating insertion at a single genomic locus, and founder and offspring showed similar levels of human IgG in the serum (Fig. 2a,b) and reproducible phenotype (Figs. 3 and 4). In contrast to human IgG1 transgenic mice[10], no hybrid (human/pig) IgG1 molecules were detected in the transgenic minipigs. A double antibody sandwich ELISA capturing the human Igκ LC and detecting human Ig HC demonstrated the expression of fully human IgG (Extended Data Fig. 1).

To characterize rearrangement of the human V gene segments in porcine B lymphocytes, messenger RNA (mRNA) samples were isolated from peripheral blood of a transgenic minipig. Sequence analysis confirmed functional V(D)J rearrangements of human variable heavy (VH) and kappa light (VK) gene segments as well as N nucleotide additions for the heavy and light chain. The latter indicates temporally coordinated rearrangement of the human heavy and light chain genes during the functioning of the porcine terminal deoxynucleotidyl transferase (TdT) responsible for N-nucleotide addition at the junction of rearranged immunoglobulins. This has also been observed in the analogous mouse model[10]. In addition, the human VH sequence contained amino acid exchanges (see Supplementary Fig. 1) due to somatic mutations outside of the complementarity-determining regions (Fig. 2c). All attempts to detect IgG4 protein in serum of hIgG transgenic minipigs were unsuccessful. However, RNA sequencing analysis revealed a small proportion of switched IgG4 genes (0.76%; Fig. 2c and Supplementary

Fig. 1). While this proves that IgG isotype switching occurs, it is a rare event and explains why no human IgG4 proteins were detected. All these indicate that the porcine and human proteins are compatible and rearrangement of human IgG genes proceeds efficiently in the transgenic minipigs.

**hIgG expression does not interfere with porcine Ab responses**

The hIgG animals are healthy and do not suffer from increased infection load. Necropsy of animals showed normal appearance of the spleen, lymph nodes and bone marrow (data not shown).

Immune competence was also confirmed experimentally using the T-cell dependent model antigen keyhole limpet haemocyanin (KLH)[16], the response to which is well-documented for Göttingen minipigs[17]. Two hIgG and two wild-type (WT) minipigs were injected subcutaneously (s.c.) with a single dose of 20 mg kg⁻¹ body weight KLH in alum adjuvant and the KLH-specific antibody response was followed over 35 d. All four minipigs mounted an IgM (not shown) and IgG response 1 week after immunization, confirming that the transgene expression does not alter T-cell dependent Ab responses to this protein (Fig. 3a). In a subsequent boosting experiment, the first dose of KLH was followed by rechallenging the animals with a second dose at day 35. The quick increase in KLH-specific porcine IgG titres after the booster immunization demonstrates that the humanized minipigs are also capable of mounting a memory response (Fig. 3b). As expected, no KLH-specific human IgG was detected (data not shown). Thus, as previously observed for hIgG1 transgenic mice[10], expression of human IgG does not compromise the immune capacity of Göttingen minipigs.

**hIgG transgenic minipigs tolerate human Abs**

Next we assessed the tolerance status of the humanized minipigs to prototypical human therapeutic Abs bevacizumab and daratumumab.

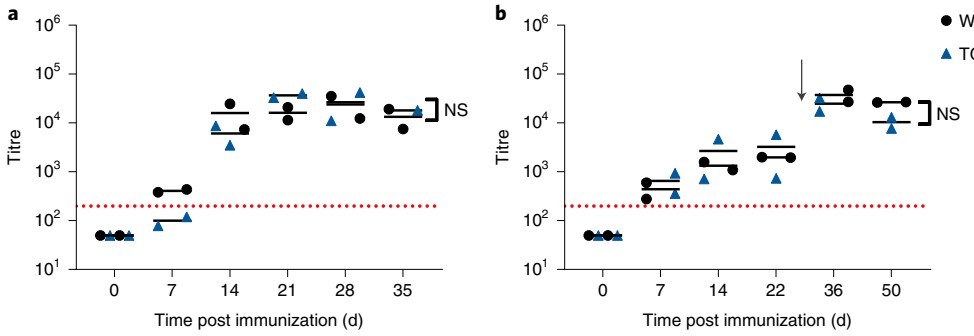

**Fig. 3 | Human IgG transgenic minipigs mount normal antibody responses towards the T-cell dependent antigen KLH. a**, Human IgG transgenic minipigs ($n = 2$) mount porcine IgG anti-KLH antibody responses at comparable levels to that of WT animals ($n = 2$) upon single immunization with KLH in alum adjuvant. **b**, Human IgG transgenic minipigs ($n = 2$) can mount memory responses after rechallenge with KLH at day 35 (arrow). The dotted red line indicates an arbitrary threshold at a titre of 200. Statistical analysis by 2-way analysis of variance (ANOVA); NS, not significant. Horizontal bar represents median. Each data point represents a biological replicate.

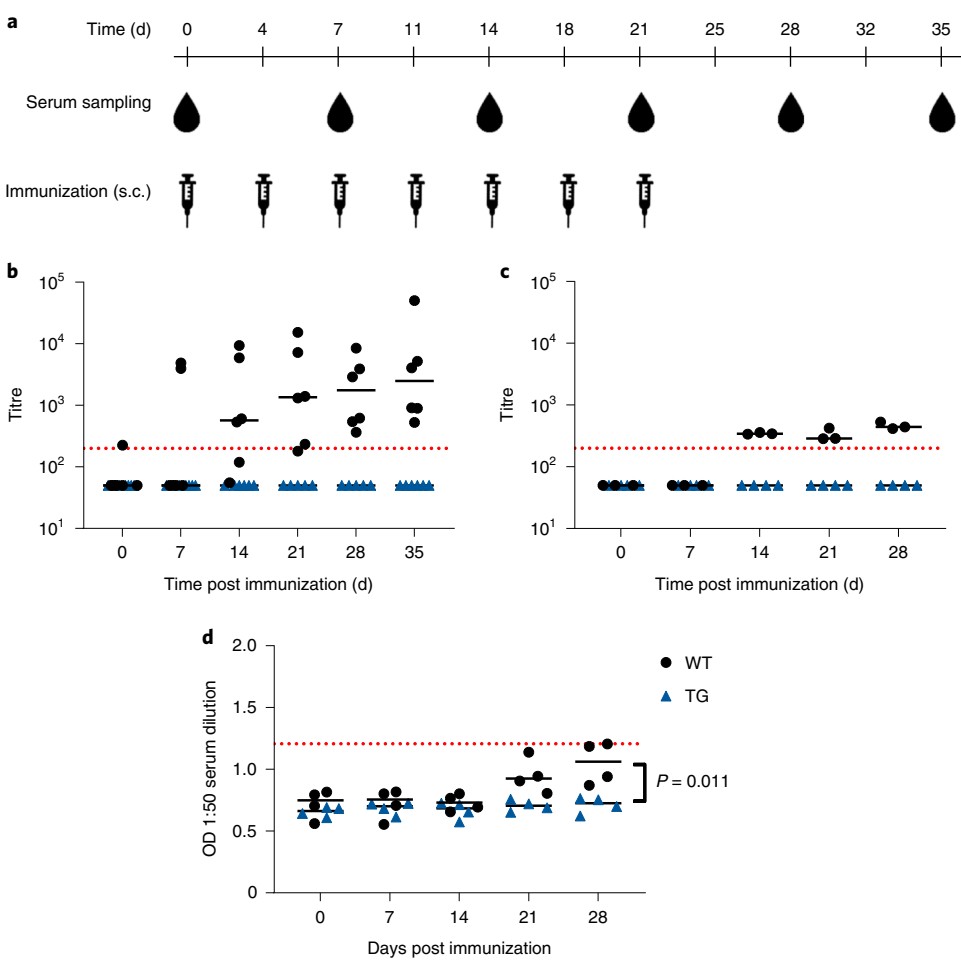

**Fig. 4 | Human IgG transgenic minipigs are tolerant to human IgG1 antibodies with a low clinical immunogenicity rate. a**, Schematic outline of the immunization strategy without adjuvant. Blood samples for ADA analysis are drawn before each treatment. **b**, Porcine IgG anti-bevacizumab antibody titre after immunization of 6 hIgG transgenic (TG) founder minipigs and 6 WT controls with bevacizumab. Data summarized from 3 individual studies, each with 2 transgenic and 2 WT minipigs. **c**, Porcine IgG anti-bevacizumab antibody titre after immunization of 4 hIgG transgenic minipigs from the F1 generation and 4 WT animals with bevacizumab. **d**, OD of porcine IgG anti-daratumumab antibodies measured via ELISA in minipigs (TG, $n = 4$; WT, $n = 4$) immunized with daratumumab. The dotted red line indicates an arbitrary threshold at a titre of 200 or mean + 6×s.d. threshold in the case of OD. Statistical analysis by 2-way ANOVA. Horizontal bar represents median. Each data point represents a biological replicate.

Both are classical antibodies of the human IgG1 isotype with low immunogenicity rates (below 1%) in patients[1].

Human IgG transgenic and wild-type minipigs were injected repeatedly with bevacizumab (0.4 mg kg$^{-1}$ body weight, 7 injections; Fig. 4a). Weekly serum samples were collected and porcine IgG ADA was measured by ELISA. Treatment with bevacizumab resulted in the formation of ADA in wild-type minipigs, but not in the hIgG-expressing founder minipigs (Fig. 4b) or their F1 offspring

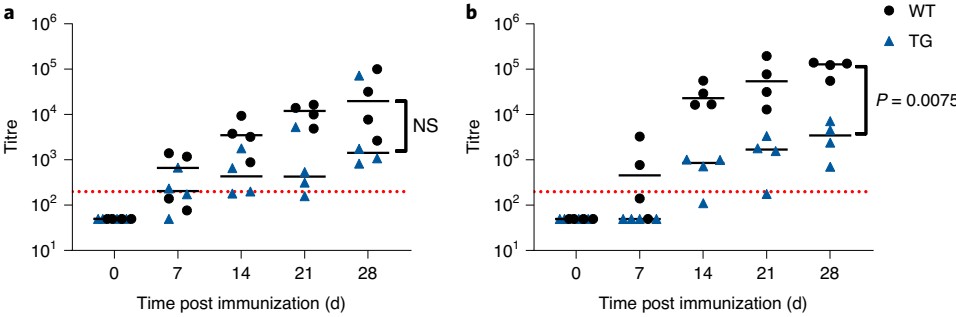

**Fig. 5 | Human IgG transgenic minipigs mount ADA responses against human IgG1 antibodies with increased clinical immunogenicity. a**, Porcine IgG anti-cergutuzumab amunaleukin antibodies after immunization (TG minipigs, $n = 4$; WT minipigs, $n = 4$) with cergutuzumab amunaleukin. **b**, Porcine IgG anti-atezolizumab antibodies after immunization with atezolizumab. The dotted red line indicates an arbitrary threshold at a titre of 200. Statistical analysis by 2-way ANOVA. Horizontal bar represents median. Each data point represents a biological replicate.

(Fig. 4c), demonstrating immune tolerance towards human IgG1 and confirming trait inheritance.

Treatment with daratumumab did not cause a substantial increase in ADA titres whether in the transgenic or in wild-type animals. However, the raw data revealed a weak but appreciable late increase in ADA signals in the wild-type animals not found in the transgenic minipig group (Fig. 4d).

All findings are in line with our previous results from IgG transgenic mice, where both Abs were also found to be non-immunogenic, while wild-type mice showed a strong (bevacizumab) or moderate (daratumumab) ADA response[10] (Supplementary Table 1).

### hIgG transgenic minipigs can assess the risk of ADA formation

To determine the potential of this model for assessing immunogenicity of therapeutic human Abs, cohorts of four hIgG and WT minipigs were treated with 0.4 mg kg$^{-1}$ body weight of atezolizumab or cergutuzumab. These are known to induce ADA responses in 39% and 70% of patients, respectively[18,19]. All minipigs developed ADA responses. There was no significant difference in ADA titre between human IgG transgenic minipigs and their wild-type littermates after treatment with cergutuzumab amunaleukin (Fig. 5a). In contrast, the ADA titre was significantly lower for the humanized minipigs after atezolizumab treatment ($P = 0.0075$) (Fig. 5b), reflecting the difference in immunogenicity between these two therapeutic Abs. The results in the human IgG transgenic minipigs recapitulate those observed in humans and data from studies in hIgG1 transgenic mice[20].

## Discussion

To ensure predictability of pharmacological studies, the ICH recommends the use of relevant animal models. However, prolonged toxicological, pharmacokinetical and pharmacodynamical studies with recombinant proteins including therapeutic antibodies are hampered by the animal's immune response against the foreign proteins. To overcome these, over the past few decades, most human recombinant proteins have been evaluated in transgenic mice tolerant to the specific human protein[21] and in the case of antibodies, in IgG-tolerant mouse models[8–10]. However, several factors have limited their application for the assessment of immunogenicity. These include the lack of genetic diversity and fundamental differences in immune mechanisms between inbred mouse strains[22,23]. Furthermore, the size of mice limits their use for studying immunogenicity via certain routes of application, such as intravitreal dosing. Due to the similarity in porcine and human anatomy, physiology and biochemistry, pigs have been suggested as a suitable non-rodent model for preclinical safety testing. Most proteins, including those of the immune system, share structural and functional similarities with their human counterparts. Compared with mice, the immune system of pigs more closely resembles that of humans[24] and they are less

inbred than mouse strains. Several lines of minipigs (Hanford, Yucatan, Yucatan micro, Sinclair and Göttingen) have been developed and are commonly used for preclinical safety assessment[25]. Göttingen minipigs have a well-defined genetic background and physiological parameters (haematology and clinical chemistry parameters, haemodynamics), making them an ideal model animal. However, it must be mentioned that pigs require higher costs compared with rodent models because more experimental reagents, animal care and husbandry are needed. Nevertheless, the size and anatomy/physiology of pigs allows more relevant pharmacokinetic studies and routes of administration, such as intravitreal injection and inhalation that are difficult to achieve in mice.

We have described a genetically engineered Göttingen minipig model for use in preclinical studies with recombinant human antibodies. By introducing a mini-repertoire of human IgG1 and IgG4 genes into the porcine germline, we have generated minipigs that are tolerant to most, although not all, human recombinant antibodies.

We have shown the successful rearrangement of human *IGH-γ1-γ4* and *IGK* germline genes and the production of serum human IgG proteins. Incorporation of human switch sequences (Sμ, Ig-Sγ1) in the transgenic construct resulted in the expression of *IgG4* mRNAs, indicating proper processing by the porcine switch machinery. The transgenic minipigs transmitted the novel traits stably to their progeny. Although expressed at low levels, the amount of human IgG protein was sufficient to induce and preserve immunological tolerance to human IgG1 Abs.

We showed immunological tolerance to human Abs by using four therapeutic Abs that either elicit (atezolizumab, cergutuzumab amunaleukin) or lack (daratumumab, bevacizumab) clinical immunogenicity. As expected, atezolizumab and cergutuzumab amunaleukin did, and bevacizumab and daratumumab did not, induce ADA in the hIgG transgenic minipigs. The immune response against bevacizumab was strong in wild-type minipigs but low after treatment with daratumumab. Daratumumab is an approved immunotherapy for multiple myeloma that depletes CD38-expressing cancer cells[26]. We cannot exclude residual binding of daratumumab to porcine CD38 and the subsequent depletion of ADA-secreting plasma cells, which may explain the low immune response. Although we have shown tolerance to a variety/number of human IgG1 antibodies, further experiments are required to assess tolerance to a much broader range of human antibodies.

The reason for ADA formation in response to atezolizumab and cergutuzumab amunaleukin antibodies is assumed to be related to their mode of action. The interleukin-2 (IL-2) variant of cergutuzumab amunaleukin cross-reacts with the IL-2 receptor of both human and pig (on the basis of sequence homology and experimental data[27]), which is known to play important roles in immunity and tolerance[28]. Atezolizumab forms close contact with human programmed death-ligand 1 (PD-L1) via 16 key amino acid residues[29], which are highly conserved among humans, mice and pigs. On the basis of the crystal structure of

the human PD-L1/atezolizumab complex[29], we generated a structural in silico model of mouse and porcine PD-L1/atezolizumab interaction that suggests stronger binding properties for the porcine complex compared with that in mice (Supplementary Fig. 2). On the basis of these data and the fact that the interaction of atezolizumab with mouse PD-L1 has been experimentally confirmed[30], similar cross-reactivity with porcine PD-L1 can be assumed. The mechanism of immunogenicity could therefore be associated with the inhibition of PD1/PD-L1 interaction, an important regulator of self-tolerance[31].

It is well known that the pharmacology of therapeutic antibodies upon binding to their targets is a major driver of toxicity and influences the immunogenicity profile[32,33]. Consequently, a lack of cross-reactivity would limit the usefulness of the hIgG transgenic animal models. As current data show a high sequence similarity between most human and porcine antigens, this should not pose a problem[34–36] but should be considered for each new antibody being tested.

The sensitivity with which hIgG transgenic minipigs respond to immunogenic compounds while tolerating non-immunogenic Abs makes them an ideal model for safety assessments of therapeutic antibodies and prediction of possible side effects.

The ICH requests that preclinical safety testing should be carried out in predictive animal models—one rodent species and one non-rodent species. The previously generated mouse lines and the newly derived humanized minipigs fulfill these requirements for human recombinant antibodies, and could therefore enable safety and efficacy testing and reduce the need for studies in NHPs.

## Methods

### Animals
Permission for the generation of transgenic pigs and the conduct of the animal experiments was issued by the government of Upper Bavaria, Germany (ROB-55.2-1-54-2532-6-13). Experiments were performed according to the German Welfare Act and European Union Normative for Care and Use of Experimental Animals.

### Constructs
Two recombinant DNA constructs encoding the soluble form of human Ig heavy chain γ (IGH) and Ig κ light chain (IGK) were generated as previously described[10]. The IGH-γ1-γ4 construct (31.4 kb) comprises a 10.9 kb region containing five variable regions (VH 1-69, VH 4-59, VH 3-30, VH 3-23, VH 1-18), five diversity regions (DH3-3, DH 4-4, DH2-8, DH3-9, DH3-10), an 8.5 kb fragment carrying joining elements JH-1 to JH-6, an intronic enhancer and switch (Sμ) region, and a 9.1 kb fragment containing constant γ-1 (exons 1–4) and γ-4 (exons 1–4) encoding the secreted IgG1 and IgG4 isoforms. All regions were flanked by endogenous sequences of human *IGH* gene between 400 bp and 1.3 kb in length. The IGK construct (10.9 kb) comprises a 1.5 kb fragment containing two variable regions (Vκ 3-20 and Vκ 1-17), a 5.6 kb fragment containing five joining elements Jκ-1 to Jκ-5, a C region and a mouse Igκ 3' enhancer (E) element. All fragments were flanked by endogenous sequences of human *IGK* gene between 500 bp and 1.7 kb in length.

### Cell culture
Porcine kidney fibroblasts (PKFs) were isolated from a male Göttingen minipig by standard method[37]. PKFs were cultured with high-glucose DMEM supplemented with 10% FBS (fetal bovine serum), 2 mM NEAA (non-essential amino acids) and 2 mM L-glutamine. Cells were passaged every 3–5 d and maintained at 50–90% confluency in an incubator at 37 °C and 5% $CO_2$. All chemicals were purchased from Sigma-Aldrich.

### Cell transfection
Passaged 1–2 PKFs were electroporated (Eporator, Eppendorf) with IGH-γ1-γ4 (31.4 kb) and IGK (10.9 kb) expression vectors and a selectable marker gene (PGK-driven *blasticidin*, BS; 1.05 kb) at a concentration of 13 μg, 4.5 μg and 0.05 μg, respectively (molar ratio of 10:10:1). All three transgenes were prepared for transfection by removing the plasmid backbone. The IGK and IGH-γ1-γ4 constructs were purified by preparative pulsed-field electrophoresis and electroelution, and the BS expression cassette by agarose gel electrophoresis using the Wizard kit (Sigma-Aldrich). For transfection, cells were subjected to an electric pulse of 1,200 V for 85 μs, followed by incubation at room temperature for 5 min. At 48 h post transfection, cells were selected with 8 μg ml$^{-1}$ BS. Individual stable transfected cell clones were isolated, samples of each clone cryopreserved at an early stage, and replicate samples cultured for further analyses.

### PCR analysis of PKF clones
DNA isolated from individual stable transfected cell clones was used for PCR screening. The following primers were used to identify the presence of IGH-γ1-γ4 transgene: IGH_F and IGH_R to amplify the 1.4 kb fragment, IGK transgene: IGK_F and IGK_R to amplify the 1.26 kb fragment. For BS transgene: BS_F and BS_R primers were used to amplify the 0.5 kb fragment. Primer sequences are given in Supplementary Table 2. PCR was carried out using the Go*Taq* polymerase. Thermal cycling parameters were: 5 min, 95 °C; then 35 cycles of: 15 s, 95 °C; 30 s, 60 °C; 1 min, 72 °C; followed by 5 min, 72 °C.

### Somatic cell nuclear transfer
Nuclear transfer was performed as previously described[38]. Briefly, in vitro matured oocytes were enucleated and a single donor cell was placed into the perivitelline space of each oocyte. After fusion and embryo activation, reconstituted embryos were transferred into the oviduct of hormonally cycle-synchronized recipient gilts. Between 80–120 reconstructed embryos were transferred to each recipient gilt.

### Determination of the transgene copy number by droplet digital PCR (ddPCR)
ddPCR was performed as previously described[39]. The transgene copy number was determined using fluorescence-labelled probes: IGH 5′FAM-ATGGGCACGACCGACCTGAGC-BHQ3′ (primers: dIGH_F and dhIGH_R); IGK 5′FAM-AGGGCTTCACAGATAGAGCTCATTTT-BHQ3′ (primers: dIGK_F and dIGK_R). GAPDH probe 5′HEX-TGTGATCAAGTCTGGTGCCC-BHQ3′ (dGAPDH_F and ddGAPDH_R) was used as reference. Primer sequences are given in Supplementary Table 2. The target genes were quantified by using the QX100 system (BioRad Laboratories).

### RNA extraction and VDJ sequencing
The blood of a 1-yr-old (F3) male human IgG transgenic minipig was collected in K2EDTA tubes and subsequently mixed with RNAlater and stored at −80 °C. RNA from blood was extracted using the RiboPure blood RNA purification kit (Invitrogen). The RNA integrity number (RIN) was >9 as determined by the Agilent 2100 Bioanalyzer. The RNA was transcribed using SuperScript IV VILO Master Mix (Invitrogen) with 500 ng total RNA as input. Subsequently, the recombined human IgG transcripts were amplified by PCR using a Q5 High-Fidelity PCR kit (NEB). First, seven forward primers specific to different variable heavy and variable kappa light chain segments were used in combination with reverse primers binding to common constant heavy and constant kappa light chain sequences. Next, PCR products were purified using QIAquick PCR purification kit (Qiagen). The PCR products from the heavy chain reactions were used as a template for nested PCR with another set of five primer pairs. All primer sequences are given in Supplementary Table 2. Subsequently, Illumina sequencing libraries were prepared from 10 pM of each purified PCR product using TruSeq Nano DNA sample preparation protocol (Illumina). The sample libraries were sequenced on an Illumina MiSeq run using paired-end sequencing for 2 × 300 cycles.

Both mates of overlapping paired-end reads were merged using usearch tool version 0.667_i86linux32 and the parameters -fastq_pctid

75 and -fastq_maxdiffs 25[40]. Subsequently, reads were translated into amino acid sequences in the anticipated frame and filtered for the presence of 5 expected adjacent amino acids and the absence of stop codons to obtain potentially functional rearrangements. Supplementary Fig. 1a depicts obtained read numbers.

## Immunoprecipitation and western blotting

MyOne Streptavidin T1 Dynabeads (Invitrogen) were washed and coated with either 4 µg biotinylated mouse anti-human IgG1 (clone HP6069, Invitrogen) or 5 µg biotinylated mouse anti-human Ig κ (clone G20-193, BD) per sample according to standard protocols. For immunoprecipitation, the coated beads were incubated overnight at 4 °C on a rotator with 100 µl minipig serum diluted 1:2 in PBS. After subsequent washing steps, the precipitated human IgG1 or human Ig κ antibodies were eluted by heating to 95 °C for 10 min with 2x Laemmli sample buffer (BioRad) and loaded directly onto mini-PROTEAN TGX gels (4–20%, BioRad). Wild-type minipig serum spiked with 100 ng human IgG1κ antibody served as a positive control and SeeBlue Plus2 (Invitrogen) together with MagicMark XP (Invitrogen) served as a molecular weight marker. After separation by electrophoresis, samples were transferred to nitrocellulose membranes (iBlot, Invitrogen). After blocking with 5% skim milk powder in tris-buffered saline-Tween (TBST), membranes were probed with 1:5,000 horseradish peroxidase (HRP) goat anti-human IgG H+L (polyclonal, Jackson) or 1:10,000 HRP mouse anti-human Ig κ (clone EPR5367-8, Abcam) and developed using SuperSignal West Femto substrate (Thermo Fisher).[41,42]

## ELISA

For the quantification of human IgG, Nunc MaxiSorp ELISA plates (Invitrogen) were coated with 1 µg ml⁻¹ of rabbit anti-human IgG H+L (polyclonal, OriGene) in coating buffer (0.1 M sodium bicarbonate pH 9.6, Alfa) at 4 °C overnight. The plates were washed three times with PBS containing 0.05% of Tween-20 before blocking with PBS containing 2% BSA (Thermo Fisher) for 1 h. Serial dilutions of minipig serum samples or purified human IgG1k as positive control were prepared in PBS + 1% FBS and incubated on the washed ELISA plates for 2.5 h at room temperature. After subsequent washing, bound human IgG was probed for 1 h at room temperature using HRP anti-human IgG (clone G18-145, BD), washed and then developed using Ultra TMB-ELISA substrate (Thermo Fisher). The reaction was stopped after 5 min using 0.18 M sulfuric acid (Merck Millipore) and the optical density (OD) was measured at 450 nm using the VersaMax ELISA plate reader. OD values of four standard curves were interpolated to calculate the concentration of human IgG1 in the serum.

To detect the presence of fully human or hybrid porcine/human IgG proteins, ELISA was performed as described above with the following antibodies: (1) coating with goat F(ab')2 anti-human κ light chain (LC) specific Ab (polyclonal, Sigma-Aldrich) and detection with an HRP-labelled anti-human IgG heavy chain (HC) Ab (clone G18-145, BD) (for recognition of human IgG); and (2) coating with mouse anti-human Ig κ LC specific Ab (clone G20-361, BD) and detection with an HRP-labelled mouse anti-pig IgG HC (clone 1G5H7, ProSci) (for recognition of porcine/human IgG).

## Immunization

Human IgG transgenic minipigs and WT Göttingen minipigs of both sexes were immunized side by side with seven subcutaneous injections twice per week of the antigen diluted in PBS. The antigens were administered in the following doses: KLH 20 mg kg⁻¹; bevacizumab, daratumumab, cergutuzumab amunaleukin, and atezolizumab 0.4 mg kg⁻¹ body weight.

Blood samples were drawn on day 0 (naïve) and at weekly intervals for 4–5 weeks before injections. Blood was collected into serum Z-Gel tubes (Sarstedt), allowed for clotting at room temperature for 30 min followed by centrifugation.

## ADA ELISA

For the detection of ADA, ELISA plates were coated with 5 µg ml⁻¹ of the antigen that was used for immunization or fragments thereof as described above. Serially diluted serum samples (1:50, then 1:3 for a total of 8 dilutions) were incubated for 2 h at room temperature and binding ADA were detected by 1 h incubation with the alkaline phosphatase-coupled AffiniPure goat anti-swine IgG (H+L) (polyclonal, Jackson) detection antibody. Subsequently, the ELISA plates were developed for 10 min with p-nitrophenyl phosphate (Merck Millipore) before reading the OD at 405 nm.

For the ADA IgG titres calculation, a cut-off value was determined for each study on the basis of the mean OD at dilution 1:50 of all naïve samples multiplied by their 6-fold standard deviation. Baseline outliers were determined as beyond 1.5× the interquartile range of quartile 3, and excluded. OD values of titrated serum samples were fitted using the 5th degree polynomial and titres were determined by the intersection of this curve with the determined threshold. All titres above the arbitrary threshold of 200 are determined as positive.

## Reporting summary

Further information on research design is available in the Nature Research Reporting Summary linked to this article.

## Data availability

The data supporting the results in this study are available within the paper and its Supplementary Information. Source data for the figures are provided with this paper. The raw and analysed datasets generated during the study are available from the corresponding authors on request. Source data are provided with this paper.

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

## Acknowledgements

We thank L. Petersen and R. Moser for experimental support, G. Steiner for titre calculations, and C. Klein and S. Klostermann for helpful input regarding antibody cross-reactivity. Additionally, we thank P. J. Knuckles, A. Kiialainen and U. Nelböck-Hochstetter for the initiation of VDJ sequencing, and A. Greiter-Wilke and E. M. Amen for continuous veterinarian and organizational support. This work was funded by F. Hoffmann-La Roche Ltd.

## Author contributions

A.I., T.S., A.S. and T.F. contributed to study conception and design. J.E., T.F., A.I. and A.S. wrote the paper. T.F., K.F., M.S., M.K., B.K., V.Z., E.W., J.E. and E.K. performed the experiments. J.E., T.F., A.S., A.I., F.W., M.E., R.S. and G.G. analysed and interpreted data. All authors provided active and valuable feedback on the manuscript.

## Competing interests

J.E., E.K., M.E., R.S., G.G., T.S., F.W. and A.I. are employees of F. Hoffmann-La Roche Ltd. The other authors declare no competing interests.

## Additional information

**Extended data** is available for this paper at https://doi.org/10.1038/s41551-022-00921-2.

**Correspondence and requests for materials** should be addressed to Felix Weber or Angelika Schnieke.

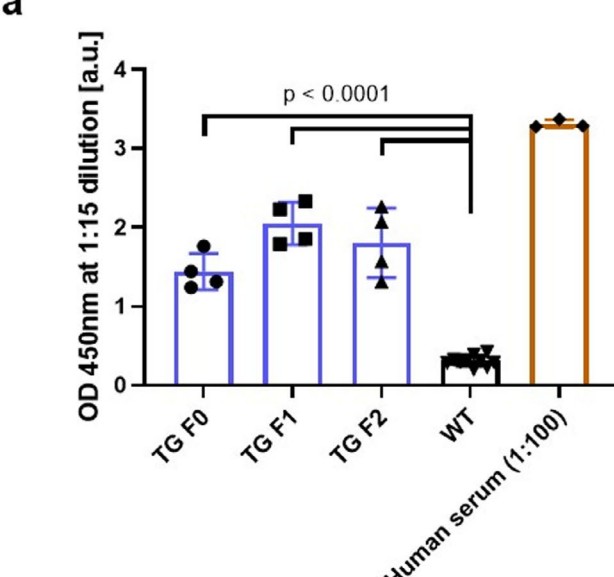

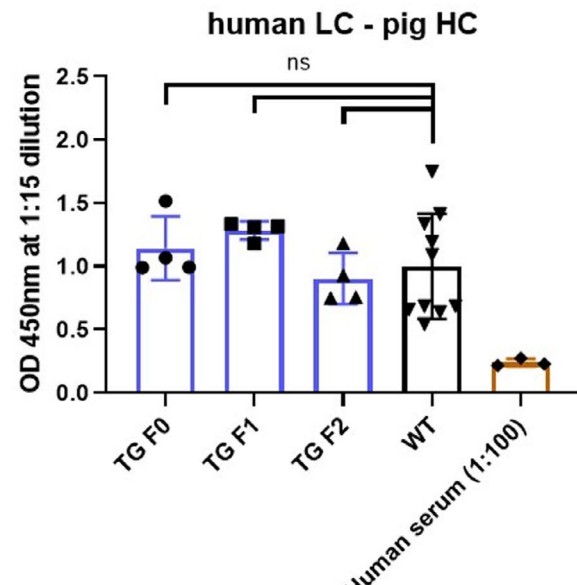

**Extended Data Fig. 1 | Fully human but no hybrid (porcine/human) IgG is detected in the serum of transgenic IgG transgenic (TG) minipig.** A double antibody sandwich ELISA was performed with serum isolated from F0 (n = 4), F1 (n = 4) and F2 (n = 4) transgenic minipigs. Human (Sigma-Aldrich) (1:100 diluted) and wild type (WT, n = 10) serum served as control samples. a, The presence of fully human IgG1 protein (coating: goat anti-human κ light chain (LC) specific Ab, detection: HRP-labelled anti-human IgG heavy chain (HC) Ab) in serum isolated from transgenic minipigs **b**, ELISA to detect hybrid porcine/human IgG (coating: mouse anti-human Ig κ LC specific Ab, detection: mouse anti-pig IgG HC) showed some cross-reactivity with porcine LC leading to a positive signal in WT and TG minipigs. However, a detection with mouse anti-pig IgG HC did not significantly increase the signal in transgenic minipigs indicating expression of fully human IgG proteins. Statistical analysis was performed by ordinary one-way analysis of variance (ANOVA) and Holm-Sidak's multiple comparisons test; p = P-value, ns = not significant, error bars represent standard deviation. Each data point represents a biological replicate.

# Reporting Summary

## Statistics

For all statistical analyses, confirm that the following items are present in the figure legend, table legend, main text, or Methods section.

| n/a | Confirmed | |
|---|---|---|
| ☐ | ☒ | The exact sample size (*n*) for each experimental group/condition, given as a discrete number and unit of measurement |
| ☐ | ☒ | A statement on whether measurements were taken from distinct samples or whether the same sample was measured repeatedly |
| ☐ | ☒ | The statistical test(s) used AND whether they are one- or two-sided<br>*Only common tests should be described solely by name; describe more complex techniques in the Methods section.* |
| ☒ | ☐ | A description of all covariates tested |
| ☒ | ☐ | A description of any assumptions or corrections, such as tests of normality and adjustment for multiple comparisons |
| ☐ | ☒ | A full description of the statistical parameters including central tendency (e.g. means) or other basic estimates (e.g. regression coefficient) AND variation (e.g. standard deviation) or associated estimates of uncertainty (e.g. confidence intervals) |
| ☒ | ☐ | For null hypothesis testing, the test statistic (e.g. *F*, *t*, *r*) with confidence intervals, effect sizes, degrees of freedom and *P* value noted<br>*Give P values as exact values whenever suitable.* |
| ☒ | ☐ | For Bayesian analysis, information on the choice of priors and Markov chain Monte Carlo settings |
| ☒ | ☐ | For hierarchical and complex designs, identification of the appropriate level for tests and full reporting of outcomes |
| ☒ | ☐ | Estimates of effect sizes (e.g. Cohen's *d*, Pearson's *r*), indicating how they were calculated |

*Our web collection on statistics for biologists contains articles on many of the points above.*

## Software and code

Policy information about availability of computer code

| | |
|---|---|
| Data collection | QX200 Droplet Generator and Reader were used for absolute quantification of target DNA. Optical density was measured at 405 nm using the VersaMax ELISA plate reader. The RNA integrity number (RIN) was determined by the Agilent 2100 Bioanalyzer. llumina sequencing libraries were prepared using Illumina TruSeq Nano DNA Sample Preparation Protocol. The sample libraries were sequenced on an Illumina MiSeq run using Paired End Sequencing for 2x300 cycles. |
| Data analysis | PCR-positive and PCR-negative droplets were counted to provide absolute quantification of target DNA in digital form using QuantaSoftTM software. OD values of standard curves were interpolated to calculate the concentration of human IgG1 in the serum. For the ADA IgG titres calculation, a cut-off value was determined for each study based on the mean OD at dilution 1:50 of all naïve samples multiplied with their 6-fold standard deviation. Baseline outliers were determined as beyond 1.5 times the interquartile range (IQR) of quartile 3, and were excluded. OD values of titrated serum samples were fitted using the 5th degree polynomial and titers were determined by the intersection of this curve with the determined threshold.<br><br>Both mates of overlapping paired end reads were merged using usearch tool version 0.667_i86linux32 and parameters -fastq_pctid 75 and -fastq_maxdiffs 25. Subsequently, reads were translated into amino acid sequences in the anticipated frame and filtered for the presence of 5 expected adjacent amino acids and the absence of stop codons to obtain potentially functional rearrangements.<br><br>Statistical analyses were performed by ordinary one-way analysis of variance (ANOVA) and Holm-Sidak's multiple comparisons test. Data were analysed using GraphPad Prism 8. |

For manuscripts utilizing custom algorithms or software that are central to the research but not yet described in published literature, software must be made available to editors and reviewers. We strongly encourage code deposition in a community repository (e.g. GitHub). See the Nature Portfolio guidelines for submitting code & software for further information.

## Data

Policy information about availability of data

All manuscripts must include a data availability statement. This statement should provide the following information, where applicable:

- Accession codes, unique identifiers, or web links for publicly available datasets
- A description of any restrictions on data availability
- For clinical datasets or third party data, please ensure that the statement adheres to our policy

The data supporting the results in this study are available within the paper and its Supplementary Information. Source data for the figures are provided with this paper. The raw and analysed datasets generated during the study are available from the corresponding authors on request.

# Field-specific reporting

Please select the one below that is the best fit for your research. If you are not sure, read the appropriate sections before making your selection.

☒ Life sciences  ☐ Behavioural & social sciences  ☐ Ecological, evolutionary & environmental sciences

For a reference copy of the document with all sections, see nature.com/documents/nr-reporting-summary-flat.pdf

# Life sciences study design

All studies must disclose on these points even when the disclosure is negative.

| Sample size | We did not conduct a formal power analysis to determine the sample size. We aimed to collect as many samples as was reasonably possible given the experimental constrains. |
|---|---|
| Data exclusions | No data were excluded from the experiments. |
| Replication | Reproducibility was tested by injecting four different monoclonal antibodies into wild-type and transgenic Göttingen mini-pigs. All ELISA measurements were performed in triplicates. |
| Randomization | The same type of material and animals were used for all experiments. Groups were formed based on the wild-type and transgenic animals. |
| Blinding | The veterinary who performed s.c. injections and collected the blood samples was blind to the statistical process. |

# Reporting for specific materials, systems and methods

We require information from authors about some types of materials, experimental systems and methods used in many studies. Here, indicate whether each material, system or method listed is relevant to your study. If you are not sure if a list item applies to your research, read the appropriate section before selecting a response.

## Materials & experimental systems

| n/a | Involved in the study |
|---|---|
| ☐ | ☒ Antibodies |
| ☒ | ☐ Eukaryotic cell lines |
| ☒ | ☐ Palaeontology and archaeology |
| ☐ | ☒ Animals and other organisms |
| ☒ | ☐ Human research participants |
| ☒ | ☐ Clinical data |
| ☒ | ☐ Dual use research of concern |

## Methods

| n/a | Involved in the study |
|---|---|
| ☒ | ☐ ChIP-seq |
| ☒ | ☐ Flow cytometry |
| ☒ | ☐ MRI-based neuroimaging |

## Antibodies

| Antibodies used | Mouse anti-Human IgG1 Fc Secondary Antibody, Biotin (Invitrogen #10467318), Biotin Mouse Anti-Human Ig κ Light Chain (BD Biosciences #555790), AffiniPure Goat Anti-Human IgG (H+L) (Jackson ImmunoResearch #109-005-003), Recombinant Anti-Kappa light chain antibody (Abcam ab124727), Rabbit anti huIgG H+L (OriGene #R1364HRP), HRP anti-human IgG (clone G18-145, Thermofisher #15838438), AffiniPure goat anti-swine IgG (H+L) (Jackson ImmunoResearch 114-035-003), Goat F(ab')2 anti-human κ light chain (Sigma-Aldrich, #SAB3701289), Mouse anti-human IgG heavy chain, HRP (BD #555788), Ig, κ Light Chain, mouse anti-human (BD #565232), Mouse anti-pig IgG (ProSci #1607). |
|---|---|
| Validation | Validation of each antibody was done under standard information offered by the supplier. |

## Animals and other organisms

Policy information about studies involving animals; ARRIVE guidelines recommended for reporting animal research

| | |
|---|---|
| Laboratory animals | Adult Göttingen Minipigs of both sexes |
| Wild animals | The study did not involve wild animals. |
| Field-collected samples | The study did not involve samples collected from the field. |
| Ethics oversight | Government of Upper Bavaria, Germany (ROB-55.2-1-54-2532-6-13). The experiments were performed according to the German Welfare Act and the European Union Normative for Care and Use of Experimental Animals. |

Note that full information on the approval of the study protocol must also be provided in the manuscript.

