## [Peer Review File · Nature Biomedical Engineering]

A humanized minipig model for the toxicological testing of therapeutic recombinant antibodies

Corresponding author: Angelika Schnieke

Editorial note

This document includes relevant written communications between the manuscript's corresponding author and the editor and reviewers of the manuscript during peer review. It includes decision letters relaying any editorial points and peer-review reports, and the authors' replies to these (under 'Rebuttal' headings). The editorial decisions are signed by the manuscript's handling editor, yet the editorial team and ultimately the journal's Chief Editor share responsibility for all decisions.

Any relevant documents attached to the decision letters are referred to as **Appendix #**, and can be found appended to this document. Any information deemed confidential has been redacted or removed. Earlier versions of the manuscript are not published, yet the originally submitted version may be available as a preprint. Because of editorial edits and changes during peer review, the published title of the paper and the title mentioned in below correspondence may differ.

Correspondence

Wed 15 Sep 2021

Decision on Article nBME-21-1576

Dear Prof Schnieke,

Thank you again for submitting to *Nature Biomedical Engineering* your manuscript, "Humanized minipig model for preclinical safety assessment of therapeutic antibodies", and for your patience in waiting for the feedback. As noted in earlier e-mail correspondence, we ended up approaching 13 experts (most of whom unfortunately declined to help, owing to travel or to busy schedules). The manuscript has been seen by three experts, whose reports you will find at the end of this message.

You will see that the reviewers appreciate the work, and that they raise a few criticisms and provide useful suggestions for improvement. We hope that with further work you can address the criticisms. In particular, we would expect that a revised version of the manuscript provides:

- * As per the suggestions of all reviewers, better framing of the introduction and the discussion regarding the technological background and the utility and limitations of the humanized minipig model.
- * Data on cross-species differences in antibody affinity, as requested by Reviewer #3.
- * Extended characterization of the humanized minipig model, and thorough methodological information, as per the various relevant reviewer comments.

Also, please provide the sequence data for all DNA constructs. Any sequence data for new constructs should be made available via an appropriate repository (see our policies at <https://www.nature.com/nature-portfolio/editorial-policies/reporting-standards#availability-of-data>).

When you are ready to resubmit your manuscript, please upload the revised files, a point-by-point rebuttal to the comments from all reviewers, the reporting summary, and a cover letter that explains the mainimprovements included in the revision and responds to any points highlighted in this decision.

Please follow the following recommendations:

- * Clearly highlight any amendments to the text and figures to help the reviewers and editors find and understand the changes (yet keep in mind that excessive marking can hinder readability).
- * If you and your co-authors disagree with a criticism, provide the arguments to the reviewer (optionally, indicate the relevant points in the cover letter).
- * If a criticism or suggestion is not addressed, please indicate so in the rebuttal to the reviewer comments and explain the reason(s).
- * Consider including responses to any criticisms raised by more than one reviewer at the beginning of the rebuttal, in a section addressed to all reviewers.
- * The rebuttal should include the reviewer comments in point-by-point format (please note that we provide all reviewers will the reports as they appear at the end of this message).
- * Provide the rebuttal to the reviewer comments and the cover letter as separate files.

We hope that you will be able to resubmit the manuscript within 20 weeks from the receipt of this message. If this is the case, you will be protected against potential scooping. Otherwise, we will be happy to consider a revised manuscript as long as the significance of the work is not compromised by work published elsewhere or accepted for publication at *Nature Biomedical Engineering*.

We hope that you will find the referee reports helpful when revising the work. Please do not hesitate to contact me should you have any questions.

Best wishes,

Pep

Pep Pàmies
Chief Editor, Nature Biomedical Engineering

Reviewer #1 (Report for the authors (Required)):

The authors develop a very useful model for assessing the safety of mAb drug candidates in relation to immunogenicity and formation of ADAs. Very few options exist to test for possible immunogenicity, beyond non-human primates or human clinical trials, therefore the value of this system is immense for the therapeutic antibody drug development field. I have a number of technical and text-related comments listed below that the authors should address to further enhance the quality of their manuscript.

- The authors should add a little more context on the different strategies taken by their humanization approach in minipigs versus what has been established in mice. Most notably, it seems the authors are using human constant regions Cgamma1 and Cgamma4, as opposed to endogenous constant regions, which is common for mice.

This comparison/contrast could go into the intro or also into the first section of results.

-Another critical point of clarification that the authors should point out in the intro is that it seems the main purpose of their humanized minipig model is for safety / immunogenicity studies of human mAb drug candidates. The main and overwhelming purpose of humanized antibody mice is for mAb drug discovery, and not safety studies. Therefore the authors should clearly denote this difference in the introduction as otherwise readers could come across confused by this fact. I may even suggest the authors consider a new

Fig. 1 that is a schematic that really points out the applications of their model, this would go a long way for clearly communicating with the readership as well.

Results, line 80-88: Can the authors comment on where genomic integration of their constructs occurred?

Since the Porcine antibody repertoire and expression remains intact, have the authors investigated whether human VL chains can be mispaired and expressed with porcine VH. Also vice-versa, do porcine VL pair with human VH?

Fig. 2 c: Potential mistake in the illustration: Vk-3-23 should be Vk-3-20 ?

Could the authors provide information if VH arrangement was also observed for VH1-69, VH4-59, VH3-23 and VH1-18?

Fig 3, the authors show antibodies serum titers following immunization with KLH. It is my understanding that these are porcine IgG titers? Did the authors test if there are human IgG titers also against KLH?

The authors should mention that the model still has limitations, since it does only include a restricted set and is missing several constant chains (IGHG2 or IgLC which are common in therapeutic antibodies) and many heavy and light chain variable genes.

T

he authors should comment on why they did not include the constant heavy chain IGHG2 or the constant light chain lambda IGLC?

The authors mentioned that IGHG4 expression could not be observed on a protein level. Is it possible to investigate further why no protein expression was observed? Did switching happen on the genomic levels or was it possible to detect IGHG4 mRNA?

The authors have to provide full annotated sequence data of all used constructs (Genbank format)

The authors have to provide more detailed information in the methods section. Which electroporation device was used? Which electroporation program? Which electroporation conditions...cell number and amount of DNA?

Reviewer #2 (Report for the authors (Required)):

This manuscript by Flisikowska et al describes the partial humanisation of the Gottingen minipig by inserting selected unrearranged human antibody V, D, J and constant regions into the genome allowing the pig to endogenously express intact secreted human IgG/K. This natural expression allows the pig to recognise human Ig as 'self' thus removing the species barrier to test human monoclonal antibody therapeutics.

Pigs (minipigs and large production pigs) have recently proven to be excellent models to predict human disease outcome, vaccine efficacy testing, vaccine dosing and delivery, pathogen epitope discovery and measuring responses to monoclonal antibody therapeutics. Anti-human antibody responses do occur and therefore this model does have potential to improve therapeutic antibody immunogenicity testing. Providing another alternative to the use of NHP is very positive for a multitude of reasons. The utility of the model is very specific to testing monoclonal antibodies, as the authors appear well aware.

The methods for genome manipulation are now well established in pigs and the humanisation strategy was based on a previous successful humanisation of mice. The fundamental principles and technical barriers has therefore been well established and overcome. Based on the pig emerging as a very powerful model, the strategy applied was relatively straight forward with clear predicted outcomes. As such, the model is clearly translatable based on the immunogenicity testing of four well characterised human therapeutic monoclonal antibodies.

Major questions (mandatory)

1. More details of the equivalent humanised mouse model is required to confirm that this model is widely used and how the immunogenicity profiles of the tested therapeutic antibodies compare between the

minipigs. What would the requirements be from this point to enable evidence generated in this model to be used to support a product for human use?

2. The disadvantages of using a larger animal and how it could be more widely applied and made available are not discussed. Access to these pigs is a key part of their true translational potential.

3. The alternative approach of making recombinant versions of human monoclonal antibodies with pig framework and constant regions is not discussed or tested. As well as removing the barrier of access to specific animals, these methods are growing in sophistication and have been applied successfully to several model and non-model species. The value of these animals must be considered against this alternative.

Minor questions (mandatory)

1. Fig2c and the data behind it is key to demonstrating the success of this model. Based on the methods this is far more comprehensive and undertaken with NGS. More quantification needs to be reported in terms of numbers and types of rearrangements. What would be predicted when looking at humans or indeed the mouse model?

2. The brief description of the human antibodies tested needs to be equivalent between each to inform the reason behind their selection. Why were these four selected?

(Non Mandatory)

3. Although not important to the model, I was surprised that there was no discussion regarding the failure of the switch sequence.

Overall a well written, comprehensive and interesting piece of work. The data supports the conclusions well.

Reviewer #3 (Report for the authors (Required)):

General comments

The authors have generated a transgenic Göttingen Minipig model carrying a mini repertoire of human Ig-gamma1 genes. These humanised pigs are said to show tolerance to all human recombinant antibodies tested. While it is a laudable approach to find replacement for non human primates, the translational relevance is somewhat hampered by inadequate discussion how often such a model may be used for monoclonal antibodies, given that many antibodies are species specific and the homology of the target molecules between primates and pigs may not be high enough.

Major comments

1) To be useful for preclinical research, it is not enough that the antibodies are tolerated and safe in guinea pigs but also effective. Biologicals have very limited toxicity and the observed safety signals are typically due to exaggerated pharmacology. The authors should therefore provide a comprehensive list of all marketed antibodies comparing the cross-reactivity to minipigs and other species, which is best included as a supplementary table. Such information can be retrieved from the European public assessment reports or FDA documents.

2) In a 2nd table the authors should provide data on species differences in the affinity between those antibodies which cross-react to minipigs.

3) I am missing a critical discussion and limitations section.

Specific comments.

1) What was the rationale for the doses and dose intervals given ?
For example the dose given to minipigs is only 0.4 mg/kg instead of 16 mg/kg.

2) What is the cross reactivity of daratumumab and the relative affinity for the pig target as compared to the human target.

3) The reasons for choosing those particular antibodies is unknown and should be provided.

- 4) The lack of ADA response to daratumumab in wild type animals has not been explained, although there is an underlying mechanism.
- 5) Why did the authors only use humanized antibodies (except for daratumumab, which prevents antibody formation) ?
- 6) Line 203 seems to be a mere speculation which should be omitted.

Mon 20 Jun 2022

Decision on Article nBME-21-1576

Dear Prof Schnieke,

Thank you for your patience in waiting for our feedback on your revised manuscript, "Humanized minipig model for preclinical safety assessment of therapeutic antibodies". As noted in previous e-mail correspondence, having consulted with the original reviewers (whose comments you will find at the end of this message), I am pleased to write that we shall be happy to publish the manuscript in *Nature Biomedical Engineering*, provided that the points specified in the attached instructions file are addressed.

When you are ready to submit the final version of your manuscript, please upload the files specified in the instructions file.

For primary research originally submitted after December 1, 2019, we encourage authors to take up transparent peer review. If you are eligible and opt in to transparent peer review, we will publish, as a single supplementary file, all the reviewer comments for all the versions of the manuscript, your rebuttal letters, and the editorial decision letters. **If you opt in to transparent peer review, in the attached file please tick the box 'I wish to participate in transparent peer review'; if you prefer not to, please tick 'I do NOT wish to participate in transparent peer review'**. In the interest of confidentiality, we allow redactions to the rebuttal letters and to the reviewer comments. If you are concerned about the release of confidential data, please indicate what specific information you would like to have removed; we cannot incorporate redactions for any other reasons.

More information on transparent peer review is available.

Best wishes,

Pep

Pep Pàmies
Chief Editor, Nature Biomedical Engineering

P.S. Nature Portfolio journals encourage authors to share their step-by-step experimental protocols on a protocol-sharing platform of their choice. Nature Portfolio's Protocol Exchange is a free-to-use and open resource for protocols; protocols deposited in Protocol Exchange are citable and can be linked from the published article. More details can be found at www.nature.com/protocolexchange/about.

Reviewer #1 (Report for the authors (Required)):

The authors have done a good revision from their initial version. My comments and suggestions for improvement have been taken into account. The new provided ELISA and RNAseq data address previous questions sufficiently. Further, the addition of more technical as well as conceptual information has improved the quality of this manuscript.

Reviewer #2 (Report for the authors (Required)):

The authors have addressed the reviewer questions comprehensively. In particular the RNA-seq and switching data. Ultimately the value of the model will need to be determined by specific applied studies.

Reviewer #3 (Report for the authors (Required)):

thanks for addressing my comments in a satisfactory manner

Nature Biomedical Engineering is a Transformative Journal. Authors may publish their research with us through the traditional subscription access route, or make their paper immediately open access through payment of an article-processing charge. More information about publication options is available.

You may need to take specific actions to comply with funder and institutional open-access mandates. If the work described in the accepted manuscript is supported by a funder that requires immediate open access (as outlined, for example, by Plan S) and your manuscript was originally submitted on or after January 1st 2021, then you will need to select the gold OA route. Authors selecting subscription publication will need to accept our standard licensing terms (including our self-archiving policies), and these will supersede any other terms that the author or any third party may assert apply to any version of the manuscript.

Rebuttal 1

Dear Madams and Sirs,

Thank you for the reviews of our manuscript, which we were pleased to see were generally very positive. Please find attached our revised version. Alterations in the text are marked in yellow. There were a number of queries raised by all reviewers and the editor. These we have answered first of all. The remaining questions in order of which they have been addressed by the reviewers.

Common points to be addressed:

1) Better framing of the introduction and the discussion regarding the technological background, applications and limitations of the humanised IgG minipigs.

Response: Following the reviewers' suggestions, we have added a new Figure (Fig. 1a) which provides a graphic summary of the experimental design and applications for human IgG transgenic minipigs. To clearly indicate that the animals were not generated for mAb drug discovery the following sentence was added: "In contrast to most antibody (Ab)-humanized mouse models for Ab discovery, our transgenic animals still express their endogenous immunoglobulin (Ig) genes, and are thus fully immune competent. The function of the human Ig transgenes is solely to induce tolerance and they do not necessarily need to be involved in an immune response." (lines 45 to 48).

Regarding the limitations of using large animal models in general the following comment has been added to the discussion:

Lines 169 to 172: "However, it has to be mentioned that pigs require higher costs compared to rodent models because more experimental reagents, animal care and husbandry are needed. On the other hand, the size and anatomy/physiology of pigs allows more relevant pharmacokinetic studies and routes of administration difficult to achieve in mice, such as intravitreal injection, inhalation."

To indicate limitations specifically due to the restricted repertoire of hIgG genes two sentences were added:

Lines: 174 to 175: "By introducing a mini-repertoire of human IgG1 and IgG4 genes into the porcine germline these minipigs are tolerant to most -although not all- human recombinant antibodies.

Lines 189 to 190: "Although we demonstrate tolerance to a variety/number of human IgG1 antibodies, further experiments are required to assess tolerance to a much broader range of human antibodies."

2) Data on cross-species differences in antibody affinity, as requested by Reviewer #3

Response: This has been done either on the basis of existing literature (daratumumab) or by species comparison of protein structure (atezolizumab). Please see discussion lines 192 to 201 and supplementary data (Suppl. Fig. 3). We also included the following sentence "It is well known that the pharmacology of therapeutic antibodies upon binding to its target(s) is a major driver of toxicity and also influences the immunogenicity profile^{32,33}. Consequently, a lack of cross-reactivity

would limit the usefulness of the hIgG transgenic animal models. As current data show a high sequence similarity between most human and porcine antigens this should not pose a problem³⁴⁻³⁶ but should be considered for each new antibody being tested" (lines 204 to 208).

3) *Extended characterisation of the humanized minipig model, and thorough methodological information, as per various relevant reviewers comments*

Response: The main request from all reviewers was to provide more details regarding the rearrangements of human IgG genes in the transgenic minipigs and to explain why human IgG4 had not been observed. To provide answers we isolated fresh blood samples and carried out in depth RNA sequence analysis (RNAseq). A detailed description of the number of sequence reads analysed, functional sequences and switch frequencies is given in Suppl. Fig. 1a. The results show the presence of functional V(D)J rearrangements of all human variable heavy (VH) and light (VK) gene segments as well as N nucleotide additions for the heavy and light chain. The newly identified rearrangements have been now included in Fig. 2c. The sequence data corresponding to the most abundant rearrangements of all human V genes are depicted in Suppl. Fig. 2. RNAseq data did not reveal any pig/human chimeric sequences. The new primer sets used for RNASeq are also included in Suppl. Table 1. RNAseq data also clearly demonstrate IgG4 isotype switch, see lines 108 to 111: "All attempts to detect IgG4 protein in serum of hIgG transgenic minipigs were unsuccessful. However, RNA sequencing analysis revealed a small proportion of switched IgG4 genes (0.76%, see Figure 2c and Supplementary Fig. 2). While this proves that IgG isotype switching occurs, it is a rare event and explains why no human IgG4 proteins were detected." A new sentence has also been added to the discussion section:

Lines 177 to 179: "Incorporation of human switch sequences (S_{μ} , Ig-S γ 1) in the transgenic construct resulted in expression of IgG4 mRNAs indicating proper processing by the porcine switch machinery."

4) *Rationale for the use of selected mAbs in the study.*

Response: The aim of this study was to show that the transgenic minipigs are tolerant towards human IgG therapeutic Abs, but at the same time mount an immune response if the human antibody was known to be immunogenic in both humans and in the previously published mouse study for species comparison. Therefore, we selected two mAb that were shown to be non-immunogenic in clinical use (daratumumab and bevacizumab) and two other Ab compounds with reported clinical immunogenicity (atezolizumab, 30%, and cergutuzumab amunaleukin, 79%). This is explained in the text (lines 129 to 130, lines 139 to 141, lines 182 to 183). For species comparison a new Table (Table 2 in Supplementary data) has been also included, showing the results obtained from the analogous mouse model. Finally, we wanted to investigate immune tolerance in the transgenic minipigs to Abs with (cergutuzumab amunaleukin, atezolizumab, bevacizumab) or without (daratumumab) cross-reactivity to pig target antigens. A short section of text and new references has been included to clarify this:

Lines 192 to 202: "The interleukin-2 (IL-2) variant of cergutuzumab amunaleukin cross-reacts with the IL-2 receptor of both human and pig (based on sequence homology and experimental data²⁷), which

is known to play important roles in immunity and tolerance²⁸. Atezolizumab forms close contact with human programmed death-ligand 1 (PD-L1) via 16 key amino acid residues²⁹, which are highly conserved between humans, mice, and pigs. Based on the crystal structure of human PD-L1/atezolizumab complex²⁹, we generated a structural model of mouse and porcine PD-L1/atezolizumab interaction, which suggests stronger binding properties of porcine complex compared to the mouse model (Supplementary Fig. 3). Since interaction of the mouse PD-L1 ortholog and atezolizumab has been experimentally confirmed³⁰, similar cross-reactivity of the porcine orthologue can also be expected. The mechanism of immunogenicity could therefore be associated with the inhibition of PD1 / PD-L1 interaction which is an important regulator of self-tolerance³¹.

In the following section we address the specific issues raised by the individual reviewers.

Review no. 1

1.1. *The authors should add a little more context on the different strategies taken by their humanization approach in minipigs versus what has been established in mice. Most notably, it seems the authors are using human constant regions Cgamma1 and Cgamma4, as opposed to endogenous constant regions, which is common for mice. This comparison/contrast could go into the intro or also into the first section of results.*

Response: As the reviewer mentions, some Ig-humanized mouse models use targeted insertion of unrearranged human V genes upstream of the corresponding mouse IgC gene segments resulting in expression of functional chimeric human V-mouse C Abs. Our approach does not alter the endogenous immune response. No functional human Abs are being produced. Expression of the soluble human IgGs has the sole purpose to induce tolerance. We have added a text to the introduction (lines 40 to 41, 45 to 48, see above point 1) stating that the main emphasis of our strategy was to ensure immune tolerance to human IgGs as opposed to obtain human mAbs. A small text change has also been made in the results section: "...as this requires the expression of the membrane bound human Ig, which is not included in the transgenic construct (see Fig. 1b)" (lines 81 to 82).

As IgG1 and IgG4 are the two Ig isotypes most commonly used in human therapeutic mAbs, we used constructs carrying unarranged human IGVH- γ 1- γ 4 (described in the text, lines 71 to 73) and IGL- κ V and J κ gene segments (described in the text, lines 77 to 78) for generation of transgenic animals. A new sentence has been added to clarify this: "Human IgG1 and hIgG4 were chosen as the two Ig isotypes most commonly used in human therapeutic mAbs¹⁶." (lines 75 to 76).

1.2. *Another critical point of clarification that the authors should point out in the intro is that it seems the main purpose of their humanized minipig model is for safety / immunogenicity studies of human mAb drug candidates. The main and overwhelming purpose of humanized antibody mice is for mAb drug discovery, and not safety studies. Therefore the authors should clearly denote this difference in the introduction as otherwise readers could come across confused by this fact. I may even suggest the authors consider a new Fig. 1 that is a*

schematic that really points out the applications of their model, this would go a long way for clearly communicating with the readership as well.

Response: We thank the reviewer for this suggestion and have included a new figure 1a. Please also see the section addressed to all reviewers, point 1.

1.3. Results, line 80-88: *Can the authors comment on where genomic integration of their constructs occurred?*

Response: We did not determine the integration site. Essential remit for the transgenic lines was a) all transgenes integrated at a single locus b) inheritance is according to Mendelian rules and c) stable expression over generations. This has been demonstrated for founder, F1 – F3 animals. The new expression data (including RNAseq) were obtained from the F3 animals, which are now being bred in the Ellegaard Göttingen Minipigs Centre (<https://minipigs.dk>). A new sentence has been included:

Lines 94 to 97: "All offspring (F1-F3) exhibited Mendelian transgenes inheritance, indicating insertion at a single genomic locus, and founder and offspring showed similar levels of human IgG in the serum (Fig. 2a, 2b) and reproducible phenotype (Fig. 3 and 4)".

1.4. Since the Porcine antibody repertoire and expression remains intact, have the authors investigated whether human VL chains can be mispaired and expressed with porcine VH. Also vice-versa, do porcine VL pair with human VH?

Response: To detect the presence of fully human or hybrid porcine/human IgG molecules, we have performed a double sandwich ELISA, which clearly demonstrated the expression of fully human IgG. The ELISA results have been included in Supplementary Fig. 1. A new sentence has also been added to clarify this:

Lines 97 to 99: „In contrast to human IgG1 transgenic mice¹⁰, no hybrid (human/pig) IgG1 molecules were detected in the transgenic minipigs. A double antibody sandwich ELISA capturing the human Igk LC and detecting human Ig HC demonstrated the expression of fully human IgG (Supplementary Fig. 1)“ The antibodies used for double sandwich ELISA have been included in the methods section (lines 312 to 317).

1.5. Fig. 2c: Potential mistake in the illustration: Vk-3-23 should be Vk-3-20?

Response: Thank you for pointing this out. We have corrected the mistake in the Illustration (Figure 1b) and replaced the designation "Vk 3-23" with "Vk 3-20".

1.6. Could the authors provide information if VH arrangement was also observed for VH1-69, VH4-59, VH3-23 and VH1-18?

Response: Yes, we showed functional rearrangements for all VH and Vk genes, including *VH1-69*, *VH4-59*, *VH3-23* and *VH1-18*. These new data are now integrated in Fig. 2c and Suppl. Fig. 2. Please see the section addressed to all reviewers, point 3.

1.7. *Fig 3, the authors show antibodies serum titers following immunization with KLH. It is my understanding that these are porcine IgG titers? Did the authors test if there are human IgG titers also against KLH?*

Response: Only the porcine endogenous IgG genes can generate functional antibodies as this requires membrane binding of Ig heavy chain. The human IgG genes only express soluble proteins and therefore cannot bind antigen on the surface of B cells. The purpose of the transgenic minipigs immunisation with KLH was to show that expression of human IgG1 does not interfere with porcine antibody response. As explained in the text (lines 117 to 118) the response to KLH in wild type Göttingen minipig is well documented. Thus, by showing that the transgenic minipigs show anti-KLH antibody responses at comparable level to WT animals, we could prove that expression of human soluble IgG does not compromise endogenous immune capacity. To address the reviewer's question directly we also tested the serum of minipigs immunized with KLH for the presence of specific human IgG using an ELISA kit from Cusabio (CSB-E16535h). All serum samples of both studies shown in the manuscript (Fig. 3a, b) were negative for human KLH-specific antibodies (IgG). For clarification we added to the text: "KLH-specific porcine IgG titers (line 124)" and "As expected, no KLH-specific human IgG was detectable (data not shown) (lines 125 to 126)."

1.8. *The authors should mention that the model still has limitations, since it does only include a restricted set and is missing several constant chains (IGHG2 or IgLC which are common in therapeutic antibodies) and many heavy and light chain variable genes.*

Response: This has been addressed. Please see the section addressed to all reviewers, point 1.

1.9. *The authors should comment on why they did not include the constant heavy chain IGHG2 or the constant light chain lambda IGLC?*

Response: As most therapeutic mAbs carry IgL kappa as Ig light chain, we chose for practical reasons not to include additional IGL lambda transgenes thus reducing the number of the transgenic constructs to be co-transfected into primary porcine cells.

1.10. *The authors mentioned that IGHG4 expression could not be observed on a protein level. Is it possible to investigate further why no protein expression was observed? Did switching happen on the genomic levels or was it possible to detect IGHG4 mRNA?*

Response: We thank the reviewer for this suggestion and indeed the new RNAseq data show a low percentage of *IGHG4* mRNA. Please see the section addressed to all reviewers, point 3.

1.11. *The authors have to provide full annotated sequence data of all used constructs (Genbank format)*

Response: The sequence of both transgenes has been submitted to the GenBank. Accession numbers have been added to the results section (lines 75 and 78).

1.12. *The authors have to provide more detailed information in the methods section. Which electroporation device was used? Which electroporation program? Which electroporation conditions...cell number and amount of DNA?*

Response: The required information was added to the method section (lines: 240, 242 and 245 to 246).

Review no. 2 – major questions

2.1. *More details of the equivalent humanised mouse model is required to confirm that this model is widely used and how the immunogenicity profiles of the tested therapeutic antibodies compare between the minipigs. What would the requirements be from this point to enable evidence generated in this model to be used to support a product for human use?*

Response: We have made substantial changes (paragraph "hIgG transgenic minipigs tolerate human Abs") to address Reviewer's comments:

Lines 133 to 135: "Treatment with bevacizumab resulted in the formation of ADA in wild type minipigs, but not in the hIgG expressing founder minipigs (Fig. 4b) or their F1 offspring (Fig. 4c), demonstrating immune tolerance towards human IgG1 and confirming trait inheritance."

Lines 136 to 138: „Treatment with daratumumab (0.4 mg/kg body weight, 7 injections; Fig. 4a) did not cause a substantial increase in ADA titers neither in the transgenic nor wildtype animals. However, the raw data revealed a weak but appreciable late increase in ADA signals in the wild type, not found in the transgenic minipig group (Fig. 4d)."

Lines 139 to 141: "All findings are in line with our previous results from IgG transgenic mice, where both Abs were also found to be non-immunogenic, while wild type mice showed a strong (bevacizumab) or moderate (daratumumab) ADA response¹⁰ (Supplementary Table 2)".

For species comparison the immunogenicity profiles of tested Ab compounds in our hIgG mouse model have been listed in Supplementary Table 2.

2.2. *The disadvantages of using a larger animal and how it could be more widely applied and made available are not discussed. Access to these pigs is a key part of their true translational potential.*

Response: For the disadvantages of using large animals we refer the reviewer to the section addressed to all reviewers, point 1. Regarding the availability and access to the hIgG transgenic

minipigs: animals have been transferred to Ellegaard Göttingen Minipigs Centre (<https://minipigs.dk>), which serves as a repository of valuable pig models for biomedical research.

2.3. *The alternative approach of making recombinant versions of human monoclonal antibodies with pig framework and constant regions is not discussed or tested. As well as removing the barrier of access to specific animals, these methods are growing in sophistication and have been applied successfully to several model and non-model species. The value of these animals must be considered against this alternative.*

Response: We agree that such "FR-porcination" would be recommended in case of non-crossreactivity to the target antigens. However, surrogate Abs require new development, production and formulation, conditions that are not always easy to achieve. Moreover, immunogenic epitopes have been shown to overlap with FR-CDR regions (Cassotta et al. 2019), calling into question the predictability of immunogenicity of "FR-porcinated" surrogate Abs. For therapeutic mAbs specific for human tumour antigens, "porcination" in the absence of antigen is negligible in healthy minipigs. However, in most cases, it would be beneficial for candidate selection to determine the intrinsic immunogenicity of the Ab skeleton in humanised Ab models. We now mention the possibility of using surrogate antibodies in the introduction (lines 40 to 42): The xeno-response "can be circumvented by using a) surrogate antibodies specific for the animal species, but their predictive value can be questioned, b) transgenic animals that express the human protein and therefore recognize it as self."

Review no. 2 – minor questions

2.4. *Fig2c and the data behind it is key to demonstrating the success of this model. Based on the methods this is far more comprehensive and undertaken with NGS. More quantification needs to be reported in terms of numbers and types of rearrangements. What would be predicted when looking at humans or indeed the mouse model?*

Response: As suggested additional experiments have been carried out. Please see the section addressed to all reviewers, point 3.

Regarding comparison to human: We refer to the study by Brezinschek et al. (1997) on the expressed repertoire in human peripheral B cells. Using a different technique (please note that the NGS technology was not available at that time), these authors established the sequence of the most frequent IGHV and IGHJ gene elements among blood B-lymphocytes. This was the knowledge we used to select the IGHV and IGHJ elements included in our transgenic constructs.

Regarding comparison to our hIgG mouse model: In both species functional rearrangements of all transgenic VH and Vk genes was detected and as well as the insertion of non-templated nucleotides (N) at the VJ joining of IGHV and IGKV genes (Fig. 2c and Suppl. Fig. 2). When compared with human we found that N(D)N regions are often -but not always- shorter in the transgenic mice and pigs (see Suppl. Fig.2). These small differences do not prevent immune tolerance to human IgG1 antibodies.

2.5. *The brief description of the human antibodies tested needs to be equivalent between each to inform the reason behind their selection. Why were these four selected? (Non Mandatory)*

Response: For answer please see the section addressed to all reviewers, point 4.

2.6. *Although not important to the model, I was surprised that there was no discussion regarding the failure of the switch sequence.*

Response: Additional RNAseq has now shown that Ig switching occurs. Please see the section addressed to all reviewers, point 3.

Review no. 3 – major comments

3.1. *To be useful for preclinical research, it is not enough that the antibodies are tolerated and safe in guinea pigs but also effective. Biologicals have very limited toxicity and the observed safety signals are typically due to exaggerated pharmacology. The authors should therefore provide a comprehensive list of all marketed antibodies comparing the cross-reactivity to minipigs and other species, which is best included as a supplementary table. Such information can be retrieved from the European public assessment reports or FDA documents.*

Response: We fully agree with the Reviewer's comment that safety issues are often due to exaggerated pharmacology rather than the structure of the compound. With the hIgG minipig model we did not intend to investigate toxicity issues, such as cytokine release syndrome or more general infusion-related reactions. We were interested in the phenomenon of ADA-dependent compound inactivation or critical exposure loss.

We addressed the question of cross-reactivity in more detail for all antibodies used in our studies and included a section in the discussion where we commented on the importance of the cross-reactivity and the probability that human mAbs may cross-react with the pig (lines 192 to 201). Additionally, we have included a Supplementary Table 2 indicating the target specificity, immunogenicity properties, and cross-species reactivity of the therapeutic antibodies used in this study.

A comprehensive list with all marketed antibodies and their cross-reactivity in the pig and other species would be a great resource, but it is beyond the remit of this publication.

3.2. *In a 2nd table the authors should provide data on species differences in the affinity between those antibodies which cross-react to minipigs.*

Response: To address the Reviewer's suggestion, we have now included Supplementary Table 2 with the required information.

Unfortunately, no affinities with the porcine target are reported in the literature for the antibodies we used in our study. The cross-reactivity of bevacizumab is based on a qualitative assessment via western blot. The binding of the IL-2 portion of cergutuzumab amunaleukin is based on sequence homology and experimental data and functional assays. In case of trastuzumab, we assume cross-reactivity to porcine PD-L1 due to sequence homology and in the case of daratumumab it is noted in the market application that this antibody does not cross-react with the pig.

We have added a more detailed explanation in the discussion about why we think that atezolizumab is cross-reactive with porcine PD-L1 (lines 194 to 200). A comparison of the PD-L1 sequences of the different species is now included in Supplementary Figure 3 and the amino acid residues critical for interaction with atezolizumab are highlighted.

3.3. I am missing a critical discussion and limitations section.

Response: This has been addressed, please see the section addressed to all reviewers, point 1.

Specific comments.

3.4. What was the rationale for the doses and dose intervals given ? For example, the dose given to minipigs is only 0.4 mg/kg instead of 16 mg/kg.

Response: The dose is based on studies with our previous human IgG1 tg mouse model (Bessa et al., 2013) where we experimentally established a time and dose regimen for immunization with human mAbs in the absence of adjuvant facilitating optimal ADA responses (Fig. 5a).

3.5 What is the cross reactivity of daratumumab and the relative affinity for the pig target as compared to the human target.

Response: Based on the EMEA report "Procedure No. EMEA/H/C/004077/0000" Daratumumab has no cross-reactivity to pig or mouse CD38. Thus, no plasma cell depletion and no pharmacological effect is expected on ADA. As indicated in Suppl. Table 2, daratumumab is moderately immunogenic in wild type mice but not in huIgG1 transgenic mice. We observed a similar tendency in the transgenic minipigs, as well (Fig. 4d). A short description and new reference have been included in the discussion: "Daratumumab is an approved immunotherapy for multiple myeloma which depletes CD38 expressing cancer cells²⁷. We cannot exclude residual binding of daratumumab to porcine CD38 and subsequent depletion of ADA secreting plasma cells, explaining the low immune response".(lines 186 to 189).

3.6 The reasons for choosing those particular antibodies is unknown and should be provided.

Response: Explanations have now been provided. Please see the section addressed to all reviewers, point 4.

3.7 *The lack of ADA response to daratumumab in wild type animals has not been explained, although there is an underlying mechanism.*

Response: We found moderate or weak ADA responses in wild type mice and minipigs, respectively (Fig. 4d and Suppl. Table 2). Assuming that daratumumab does not cross react to CD38 in these animal species, we interpret these ADA findings as reflecting intrinsic immunogenic properties of this therapeutic Ab. We then interpret the lack of ADA response to daratumumab in our transgenic mouse and minipigs (Suppl. Table 2 and Fig. 4b) as resulting from immune tolerance towards human IgG1 Abs. Nonetheless, the lack of cross-reactivity has not been investigated in detail *in vivo* such that remaining binding to CD38 cannot be totally excluded and could explain the observed low immunogenicity of this Ab compound in wild type mice and minipigs.

3.8 *Why did the authors only use humanized antibodies (except for daratumumab, which prevents antibody formation)?*

Response: Please see the section addressed to all reviewers, point 4.

3.9 *Line 203 seems to be a mere speculation which should be omitted.*

Response: We thank the reviewer for pointing out that the sentence about the cross-reactivity between atezolizumab and porcine PD-L1 was speculative. This sentence has been removed. To assess if a cross reactivity is theoretically possible *in silico* structural models for the interaction of atezolizumab with both mouse and porcine PD-L1 were analysed and are now included in a new Supplementary Fig. 3. *In silico* modelling suggests stronger binding properties of the porcine PD-L1/atezolizumab complex compared to mouse. A short description and new reference have now been included in the discussion “Atezolizumab forms close contact with human programmed death-ligand 1 (PD-L1) via 16 key amino acid residues²⁹, which are highly conserved between humans, mice, and pigs. Based on the crystal structure of human PD-L1/atezolizumab complex²⁹, we generated a structural *in silico* model of mouse and porcine PD-L1/atezolizumab interaction, which suggests stronger binding properties for the porcine complex compared to mouse (Supplementary Fig. 3). Based on this data and the fact that the interaction of atezolizumab with mouse PD-L1 has been experimentally confirmed³⁰, a similar cross-reactivity with porcine PD-L1 can be assumed (lines 194 to 200).”

We trust this has fully addressed all questions.